# Impact of COVID-19 on emergency medical services utilization and severity in the U.S. Upper Midwest

Moshe Shalom[1], Brett Boggust[2], M. Carson Rogerson, IV[3], Lucas A. Myers[4], Shuo J. Huang[4], Rozalina G. McCoy[3,4,5]*

1 Tel Aviv University Sackler School of Medicine, Tel Aviv, Israel, 2 Creighton University School of Medicine, Omaha, Nebraska, United States of America, 3 Mayo Clinic Ambulance, Rochester, Minnesota, United States of America, 4 University of Maryland Institute for Health Computing, North Bethesda, Maryland, United States of America, 5 Division of Endocrinology, Diabetes, and Nutrition, Department of Medicine, University of Maryland School of Medicine, Baltimore, Maryland, United States of America

* Rozalina.McCoy@som.umaryland.edu

**Data Availability Statement:** A de-identified dataset that includes all data necessary for study

## Abstract

The COVID-19 pandemic has claimed over one million lives in the United States and has drastically changed how patients interact with the healthcare system. Emergency medical services (EMS) are essential for emergency response, disaster preparedness, and responding to everyday emergencies. We therefore examined differences in EMS utilization and call severity in 2020 compared to trends from 2015–2019 in a large, multi-state advanced life support EMS agency serving the U.S. Upper Midwest. Specifically, we analyzed all emergency calls made to Mayo Clinic Ambulance, the sole advanced life support EMS provider serving a large area in Minnesota and Wisconsin, and compared the number of emergency calls made in 2020 to the number of calls expected based on trends from 2015–2019. We similarly compared caller demographics, call severity, and proportions of calls made for overdose/intoxication, behavioral health, and motor vehicle accidents. Subgroup analyses were performed for rural vs. urban areas. We identified 262,232 emergent EMS calls during 2015–2019 and 53,909 calls in 2020, corresponding to a decrease of 28.7% in call volume during 2020. Caller demographics shifted slightly towards older patients (mean age 59.7 [SD, 23.0] vs. 59.1 [SD, 23.7] years; p<0.001) and to rural areas (20.4% vs. 20.0%; p = 0.007). Call severity increased, with 95.3% of calls requiring transport (vs. 93.8%; p<0.001) and 1.9% resulting in death (vs. 1.6%; p<0.001). The proportion of calls for overdose/intoxication increased from 4.8% to 5.5% (p<0.001), while the proportion of calls for motor vehicle collisions decreased from 3.9% to 3.0% (p<0.001). All changes were more pronounced in urban areas. These findings underscore the extent to which the COVID-19 pandemic impacted healthcare utilization, particularly in urban areas, and suggest that patients may have delayed calling EMS with potential implications on disease severity and risk of death.

analyses is included with the submission as Supporting information files.

**Funding:** This effort was funded by the National Institute of Diabetes and Digestive and Kidney Diseases (NIDDK) of the National Institute of Health (NIH) National Institute of Diabetes and Digestive and Kidney Diseases (NIDDK) grant number R03DK127010 (RGM) and 5T32DK098107-09 (SJH). RGM is an investigator at the University of Maryland-Institute for Health Computing, which is supported by funding from Montgomery County, Maryland and The University of Maryland Strategic Partnership: MPowering the State, a formal collaboration between the University of Maryland, College Park and the University of Maryland, Baltimore.

**Competing interests:** The authors have declared that no competing interests exist.

## Introduction

The Coronavirus Disease 2019 (COVID-19) pandemic, caused by the viral agent SARS-CoV-2, has affected nearly 90 million Americans and claimed more than one million lives in the United States (U.S.) as of February 2023 [1]. COVID-19 most often presents as an acute respiratory syndrome, with higher risk of severe disease among individuals who are older, immunocompromised, or with multiple comorbid conditions [2]. High rates of COVID-19 cases, as well as mitigation efforts implemented by governments and private entities to curb the spread of the disease, have led to widespread disruptions across all sectors of life [3]. Several studies have observed decreases in hospital utilization for conditions other than COVID-19 infection throughout the pandemic, particularly during peaks of case incidence [4–6]. Health systems' deferral of elective surgical procedures and preventive medical encounters early during the pandemic, as well as patient avoidance of hospitals and clinics due to fear of contracting COVID-19 or concerns about inadequate capacity of the healthcare system to address less urgent concerns, likely contributed to these observed decreases [4, 5]. As a result, mortality due to both COVID-19 and unrelated causes increased during the pandemic, particularly during its first year for which robust population-level data are now available [7]. Emergency medical services (EMS) are a core component of the U.S. healthcare system and serve as a first line of response in times of crisis. However, data on changes in EMS utilization, including call volume and acuity, during the pandemic are limited. This is particularly true in rural areas, where reliance on EMS may be greater and availability of alternative sources of healthcare may be more limited.

EMS plays an important role in the medical care of acutely ill and injured patients. For high acuity calls, such as acute myocardial infarction, stroke, and trauma, EMS transport to the emergency department (ED) is associated with reduced time to treatment, timely receipt of definitive therapy and reduced mortality [8–13]. Two studies described the impact of COVID-19 on EMS utilization, suggesting that call volumes may have decreased early in the pandemic [14, 15]. One study, conducted in Western Pennsylvania in the spring of 2020 when COVID-19 first emerged in the U.S., showed that there was a significant decrease in EMS response, but 911 callers comprised an overall sicker patient population than before the pandemic as gauged by greater prevalence of abnormal vital signs. However, there was also an increase in nontransport cases, suggesting that lower acuity calls may have also increased [15]. A French study conducted during the same early period of the pandemic similarly found an overall decrease in calls, but the number of calls related to infection, chest pain, and breathing difficulty increased [14]. However, data on EMS call volumes during later periods of the pandemic remain scarce. Moreover, prior studies have not focused on rural areas of the U.S., where EMS utilization patterns, patient populations, and response to the COVID-19 pandemic may differ from the more studied urban areas.

With the COVID-19 pandemic now mostly behind us, it is important to understand the impact of the pandemic—in its different phases—on EMS utilization. Leveraging data from a large, advanced life support (ALS) ambulance organization serving rural and urban communities across two states in the U.S. Upper Midwest, we compare the volumes and severity of EMS calls during 2020 to those at a baseline period between 2015–2019. We further probe for potential differences in call volume and severity as a function of rurality, as there is evidence of differences in COVID-19 prevalence between urban and rural communities [16, 17]. We hypothesized that EMS call volumes have decreased, while severity increased, reflecting the general public's underlying avoidance of hospitals and deferral of ambulatory care. Furthermore, we hypothesized that calls to EMS in rural areas decreased to a lesser extent than in urban areas, as rural communities are often more reliant on EMS than urban communities.

## Methods

### Study design

This cross-sectional study was conducted using the Patient Care Record (PCR) of Mayo Clinic Ambulance. The study was conducted and is reported in accordance with STROBE guidelines for observational cohort studies [18].

### Ethics

The study was exempt from review by the Mayo Clinic Institutional Review Board as it involved analysis of de-identified data. Need for consent was waived as the study used de-identified data and obtaining consent would be infeasible.

### Study population and setting

We used Mayo Clinic Ambulance PCR to identify all EMS encounters between January 1, 2015 and December 31, 2020, subsetting this period to 2015–2019 (baseline, pre-pandemic period) and 2020. First cases of COVID-19 were detected in Wisconsin on February 5, 2020 and in Minnesota on March 6, 2020 [19]. Stay-at-home orders started on March 25 and March 28, respectively, while schools closed on March 18 in both states [19]. COVID-19 deaths first began to peak in early October 2020 in Wisconsin, while in Minnesota there was a smaller mortality spike in May 2020 followed by a larger peak in November/December 2020 [20].

Included EMS encounters were 911 activations, patient presentations to an EMS station (i.e, "walk-ups"), and requests for service from other public safety agencies such as law enforcement and fire agencies. Non-emergent requests for service (i.e., interfacility transports) were excluded.

Mayo Clinic Ambulance is an ALS provider and the primary response, treatment, and ground and air medical transport service in communities across southern, central, and western Minnesota (hubs in Albert Lea, Austin, Duluth, Fairmont, Litchfield, Little Falls, Mankato, Owatonna, Plainview, Rochester, and St. Cloud) as well as western Wisconsin (hubs in Barron, Eau Claire, Osseo, and Superior), covering 6,894 square miles of urban, suburban, and rural areas. Mayo Clinic Ambulance is staffed by paramedics and emergency medical technicians, and responds to approximately 100,000 requests for service annually, including 75,000 emergent 911 calls, excluding calls for inter-facility transport. In Minnesota and Wisconsin, a single ambulance service is licensed to provide emergency care to an assigned primary service area with defined geographic boundaries.

### Outcome variables

EMS activations were categorized based on outcome, severity, primary impression, and location using structured data in the PCR. Each call had two possible outcomes: transport to the ED vs. no transport. If the patient was transported, the urgency of transport was also categorized, with lights and sirens use as one measure of perceived urgency. To assess severity, calls were categorized as (1) no transport, which includes both 'treat, no transport' and 'no treatment or transport needed' as in some situations these two categories can be used interchangeably; (2) refusal of transport against medical advice (AMA); (3) treat, routine transport; (4) treat, transport with lights and sirens; and (5) any call with death prior to ED arrival. High call severity was also gauged by requirement for cardiopulmonary resuscitation (CPR) while on scene or during transport.

The decision to use lights and sirens is made by the EMS provider based on a combination of the patient care guideline (PCG) developed and maintained by Mayo Clinic Ambulance

medical direction, as well as the EMS provider's clinical judgement. The PCG advises that, at the discretion of the ambulance crew, transport with emergency lights and sirens may be considered if the following clinical conditions or circumstances exist: (1) difficulty in addressing issues related to airway, breathing or circulation; (2) severe trauma; (3) severe neurological or cardiac conditions; (4) obstetrical emergencies; or (5) for patients who pose a safety threat to themselves or the crew after reasonable attempts to control the situation or the patient have been attempted and failed. EMS providers are encouraged to minimize use of lights and sirens to enhance safety for the patient, caregivers, passengers, and general community, and are required to document the rationale for using lights and sirens in the PCR when they have done so. Thus, the decision to use lights and sirens is influenced by the patient's clinical situation, distance to the ED and traffic conditions while en route, and the EMS providers' experience and resources available at the time of the call.

We also queried the primary impression of the call as documented by the EMS provider in the PCR at the conclusion of the call. Our primary analyses considered all calls, while secondary analyses focused specifically on calls with the primary impressions of motor vehicle collision, overdose or intoxication, and behavioral health. These subgroup analyses were chosen based on evidence that these COVID-19-unrelated events as well as deaths for these conditions may have been influenced by the COVID-19 pandemic [7, 21–25].

## Independent variables

Patient demographics were ascertained from the PCR at the time of each call and included patient age, gender, and address of the pick-up location (to assign state and determine rurality). Call location was categorized as rural vs. urban based on the pick-up location documented in the PCR. Rural status was categorized as rural vs. urban using rural-urban commuting area (RUCA) codes, where metropolitan areas were classified as urban, while micropolitan areas, small towns, and rural areas were classified as rural [26–28]. Race and ethnicity data are not routinely collected during calls and were therefore not included. Missing values for each variable are presented as a separate category ('unknown') and included in the analyses; the rate of missing data was <1% for all included variables.

## Statistical analyses

Baseline patient characteristics were descriptively summarized at the call level and compared between the 2015–2019 and 2020 periods using the Chi Square test or the Wilcoxon two-sample test, as appropriate. Descriptive data for each quarter of 2020 were compared to the average of the same quarter of the previous 5 years (2015–2019).Calendar quarters were defined as three-month periods beginning January 1 of a given calendar year. This approach was chosen to standardize time periods of outcome ascertainment across the two states included in our analysis, which had different COVID-19 case and fatality trajectories and policy approaches to risk mitigation. Two-sided p-values of <0.05 were considered to be statistically significant.

To visualize the geographic distribution of EMS call volumes between 2015 and 2020, we mapped the change, by county, in the annual EMS call rate per 10,000 county residents from 2015–2019 to 2020 periods using QGIS 3.34.2. We averaged 2015–2019 call volumes to estimate a one-year average for the period. To calculate each county's per resident call rate, we drew county population estimates from US Census's 2020 decennial census data. We restricted our map to Minnesota and Wisconsin using OpenStreetMaps as a base layer.

Call volume was modeled as a function of time using simple linear regression with call volumes between January 1, 2015 and February 28, 2020. The dependent variable in this model was the number of 911 calls which occurred in a calendar month, and the independent variable

was the ordinal number of the month in the progression between the start and finish of this timeframe. Coefficients from this model were used to calculate the expected call volume after March 1, 2020. Goodness of fit for the model was assessed using the $R^2$, Root MSE, and ANOVA statistics. We then compared expected call volumes with those observed by calculating the absolute difference between expected and observed call numbers (expected/modeled volume minus observed volume) and the relative difference between the two.

Overall and for each calendar year quarter of the 2015–2019 and 2020 periods, we compared the difference in the distribution of call outcomes, severity, rurality, and primary impressions using the Chi Square test. Bonferroni correction for multiple comparisons was used to reduce the probability of type 1 error (false positive), adjusting the p-value level that would signify statistical significance after correction to.

All data management and analyses were carried out using SAS 9.4 (SAS Institute Inc. Cary, NC).

## Results

There were 262,232 EMS activations between 2015–2019 (corresponding to an average of 52,446.4 per year) and 53,909 in 2020. While statistically significant, the overall changes in the demographics of patients served in 2020 compared to 2015–2019 were minor (Table 1). Mean age of patients calling EMS in 2020 was 59.72 (± 23.71), compared to 59.05 (± 23.71) years in 2015–2019 (p<0.001), and a slightly greater proportion was male (49.57% vs. 47.94%; p<0.001).

The mean number of calls per month over the whole study period was 4,391 (SD, 287.1). Using data from 2015–2019, we modeled the expected call volumes for 2020; $R^2$ = 0.6463. Considering the anticipated changes in call volumes over time, we observed a relative call volume decrease of 5.3% in 2020 compared to 2015–2019, from an expected call volume of 4744.5 per month to an observed 4492.4 calls per month. This decrease was more pronounced in urban

**Table 1. Study population.** Description of EMS calls included in this study, subset based on study period as occurring between 2015–2019 (the baseline pre-COVID-19 period) or 2020.

|  | 2015–2019 | 2020 | p-value |
|---|---|---|---|
| **Total call number** | 262,232 | 53,909 | |
| **Patient age, years, mean (SD)** | 59.05 (23.71) | 59.72 (22.98) | <0.001 |
| **Patient age, years, category, N (%)** | | | |
| <18 | 12,416 (4.73%) | 2,081 (3.86%) | <0.001 |
| 18–29 | 26,175 (9.98%) | 4,896 (9.08%) | <0.001 |
| 30–44 | 32,813 (12.51%) | 7,209 (13.37%) | <0.001 |
| 45–59 | 47,534 (18.13%) | 9,380 (17.40%) | <0.001 |
| 60–74 | 60,641 (23.12%) | 13,369 (24.80%) | <0.001 |
| ≥75 | 81,364 (31.03%) | 16,725 (31.02%) | 0.96 |
| Unknown | 1,289 (0.49%) | 249 (0.46%) | |
| **Gender, N (%)** | | | |
| Women | 134,256 (51.20%) | 26,780 (49.68%) | <0.001 |
| Men | 125,709 (47.94%) | 26,724 (47.57%) | |
| Other or unknown | 2,267 (0.86%) | 405 (0.75%) | |
| **Rurality of location, N (%)** | | | |
| Rural | 52,342 (19.96%) | 11,021 (20.44%) | 0.007 |
| Urban | 209,451 (79.87%) | 42,733 (79.27%) | |
| Unknown | 439 (0.17%) | 155 (0.29%) | |

areas (6.8% decrease, from an expected volume of 45,856 calls to 42,733 observed calls) compared to rural areas (0.05% increase, from an expected volume of 11,016 calls to 11,021 observed calls) (Fig 1). Call volumes declined the most early in the COVID-19 study period, decreasing 28.71% in April 2020 and 14.83% in May 2020. The geographic distribution of calls across counties comprisingthe Mayo Clinic Ambulance primary service areas in the 2015–2019 and 2020 time periods is depicted in Fig 2.

While the overall observed number of EMS activations decreased during 2020 compared to 2015–2019, the proportion of observed calls requiring transport to the ED increased from 93.77% to 95.33% (p<0.001) (Table 2). The proportion of calls with AMA refusal of transport decreased during 2020, from 1.43% to 1.20% (p<0.001). The proportion of calls requiring initiation of CPR also increased from 1.03% to 1.39% (p<0.001), while the proportion of calls involving death increased from 1.59% to 1.93% (p<0.001). The proportion of observed calls involving death was most increased above expected in Q2 (from 1.53% to 2.08%, p<0.001) and Q4 (from 1.65% to 2.08%, p<0.001) of 2020, which correspond to April through June 2020 (Q2) and October through December 2020 (Q4). Notably, a greater than expected proportion of activations were transported to the ED without lights and sirens, with lights and siren use decreasing from 4.15% to 3.01% (p<0.001).

The proportion of calls for overdose or intoxication increased during the COVID-19 period (5.46%) when compared to pre-COVID-19 period (4.84%); p<0.001 (Table 2; Fig 3). This increase was statistically significant in both urban and rural areas, though rates were higher in urban areas during both time periods. In contrast, the proportion of calls for motor vehicle collisions decreased from 3.91% during the pre-COVID-19 period to 3.00% during the COVID-19 period; p<0.001 (Table 2; Fig 2). This change was larger in urban than rural areas, though motor vehicle collisions made up a higher proportion of calls in rural than urban areas.

## Discussion

Analysis of nearly 54,000 EMS calls made in 2020 in the U.S. Upper Midwest, compared to over 262,000 calls made between 2015 and 2019, revealed that while EMS call volumes were lower in 2020 than expected based on historic trends, the severity of calls increased, including the proportion of calls with a fatal outcome. The decrease in call volumes was greater in urban than rural areas. Additionally, the proportion of calls for overdose or intoxication increased, while the proportion of calls for motor vehicle collisions decreased, reflecting the impact of the COVID-19 pandemic on different aspects of society.

Recent studies of hospital use during the COVID-19 pandemic found that hospital admissions decreased by up to 50% during the early months of the pandemic, especially during the April 2020 peak in case incidence in the Northeast region of the U.S [4, 5]. In Minnesota, all-cause mortality increased 11.8% in 2020 relative to prior years, with greatest increases in deaths from assault by firearms (68% increase), overdose (49% increase), alcoholic liver disease (26% increase), cirrhosis (28% increase), and malnutrition (48%) increase; COVID-19 comprised 9.9% of all deaths in the state [7]. Our findings build on this body of evidence by examining the impact of the pandemic on pre-hospital emergency care.

We hypothesize that there are several potential explanations for the decline in EMS activation and the concurrent increases in patient severity and adverse outcome observed in our data. Early in the pandemic, corresponding to Q2 of 2020, rates of COVID-19 infection were low in Minnesota and Wisconsin (Minnesota positive test rates did not exceed 5% until the week of May 10[th], 2020) [29]. Nevertheless, facing an emergence of a poorly understood virus and rising death rates in other areas of the U.S., hospitals suspended elective surgeries and procedures, while clinics deferred preventive visits and transitioned many other encounters to

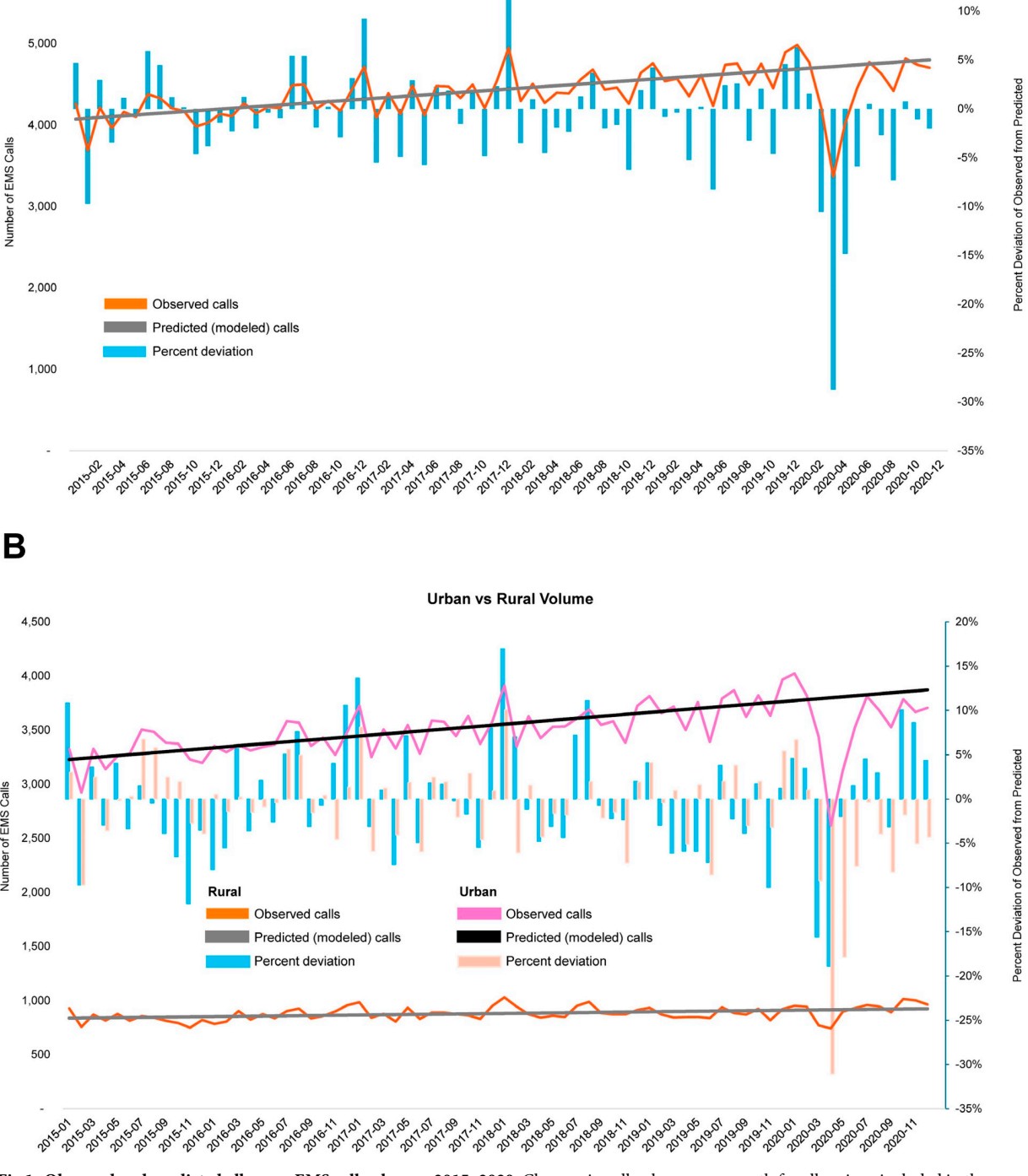

**Fig 1. Observed and predicted all-cause EMS call volumes, 2015–2020.** Changes in call volumes per month for all regions included in the study, overall (upper graph) and subset by rural status of the call originating location (bottom graph). Observed call volumes were compared to those expected based on a linear model of event rates across the study period. Percent deviation is calculated as the observed volume minus the expected (i.e., modeled) volume, divided by the expected volume.

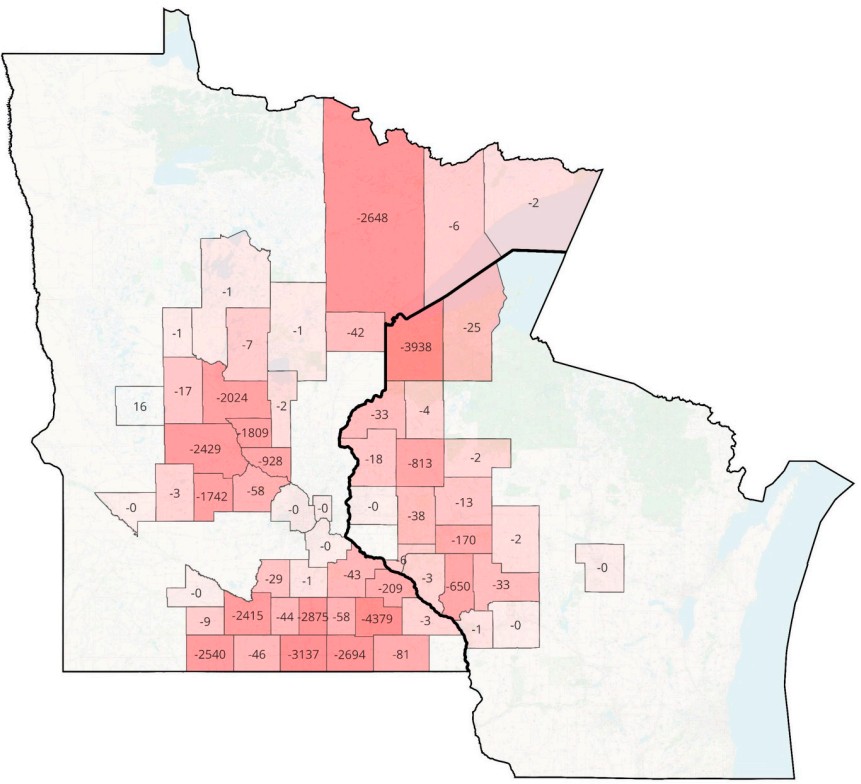

**Fig 2. County-level geographic distributions of 2020 rate changes in EMS calls across Mayo Clinic Ambulance primary service areas.** The map shows, for each county included in the Mayo Clinic primary service area in Minnesota and Wisconsin, the change in per 10,000 county resident EMS call rate in 2020 as compared to the annual average in 2015–2019, using the 2020 U.S. Census county population. Darker red shade in the color gradient represents a larger magnitude of negative call rate change.

virtual platforms. Fewer surgeries, procedures, and other interventions may have resulted in fewer acute health events that could have prompted an EMS call. Worried about the risk of exposure and heeding social distancing guidelines, people may have deferred seeking both routine and urgent care for lower acuity conditions. Thus, while there were fewer EMS activations overall, those that did occur were more likely to be for more serious conditions requiring ED transport, with some of the deferred activations being for lower acuity conditions that would not have required transport. Simultaneously, the higher proportion of activations with death as an outcome during the pandemic also suggests that at least some of the deferred calls should have been made earlier but were deferred with a fatal outcome.

Our findings contrast with other studies that examined EMS utilization in the context of the COVID-19 pandemic in other settings and populations. Lane et al examined EMS utilization in Calgary, Canada between December 2019 and June 2020 (corresponding roughly to Q1-Q2 2020 in our study), with the first case of COVID-19 reported midway through that time period [30]. They found that in 2020, as compared to 2017–2019, overall EMS call volumes increased, as did the proportion of calls that were of higher clinical severity, while overall transport rates decreased (observed mainly in the lower severity calls) [30]. Higher EMS volumes and greater proportion of calls for low-acuity needs were also observed in a New Zealand study by Dicker et al, which examined EMS utilization in March and April 2020 during a peak of infection rates and a general country lockdown period [31]. We, too, found a higher

**Table 2. Observed severity and characteristics of calls during the 2015–2019 and 2020 time periods.** All values, except for p-values, are listed as percentages of the total call volume during that quarter. Quarter 2 of 2020 (April through June 2020) corresponds to the beginning of the COVID-19 pandemic in Minnesota and Wisconsin, U.S.A. AMA, against medical advice. CPR, cardiopulmonary resuscitation. ED, emergency department.

| | Q1 | | | Q2 | | | Q3 | | | Q4 | | | Total Q1-Q4 | | |
|---|---|---|---|---|---|---|---|---|---|---|---|---|---|---|---|
| | 2015–2019 | 2020 | P[4] | 2015–2019 | 2020 | P[4] | 2015–2019 | 2020 | P[4] | 2015–2019 | 2020 | P[4] | 2015–2019 | 2020 | P |
| **Transport Category** | | | | | | | | | | | | | | | |
| Transport (vs. no transport) | 93.99% | 94.96% | <**0.001** | 93.40% | 95.46% | <**0.001** | 93.59% | 95.20% | <**0.001** | 94.11% | 95.73% | <**0.001** | 93.77% | 95.33% | <**0.001** |
| Lights & sirens (vs. without)[1] | 3.67% | 3.00% | <**0.001** | 4.18% | 2.97% | <**0.001** | 4.48% | 3.29% | <**0.001** | 4.26% | 2.79% | <**0.001** | 4.15% | 3.01% | <**0.001** |
| **Call Outcome** | | | | | | | | | | | | | | | |
| Treat, no transport | 4.60% | 3.83% | <**0.001** | 5.10% | 3.18% | <**0.001** | 4.76% | 3.08% | <**0.001** | 3.97% | 3.02% | <**0.001** | 4.6% | 3.28% | <**0.001** |
| AMA treat, no transport | 1.23% | 1.02% | .04 | 1.31% | 1.17% | .02 | 1.44% | 1.52% | .47 | 1.74% | 1.08% | <**0.001** | 1.43% | 1.20% | <**0.001** |
| Treat, routine transport | 89.49% | 90.92% | <**0.001** | 88.56% | 91.17% | <**0.001** | 88.46% | 90.92% | <**0.001** | 89.07% | 91.55% | <**0.001** | 88.90% | 91.14% | <**0.001** |
| Treat, transport with lights & sirens[2] | 3.07% | 2.43% | <**0.001** | 3.50% | 2.40% | <**0.001** | 3.75% | 2.70% | <**0.001** | 3.58% | 2.27% | <**0.001** | 3.48% | 2.45% | <**0.001** |
| Call with death prior to ED arrival | 1.60% | 1.79% | 0.10 | 1.53% | 2.08% | <**0.001** | 1.59% | 1.78% | 0.12 | 1.65% | 2.08% | <**0.001** | 1.59% | 1.93% | <**0.001** |
| **Need for CPR** | | | | | | | | | | | | | | | |
| CPR (vs. no CPR) | 1.01% | 1.29% | <**0.001** | 1.00% | 1.40% | <**0.001** | 1.04% | 1.26% | 0.02 | 1.08% | 1.62% | <**0.001** | 1.03% | 1.39% | <**0.001** |
| CPR (in rural areas) | 1.41% | 1.87% | 0.07 | 1.27% | 1.63% | 0.14 | 1.40% | 1.89% | 0.05 | 1.47% | 2.04% | 0.02 | 1.39% | 1.87% | <**0.001** |
| CPR (in urban areas) | 0.90% | 1.15% | **0.01** | 0.93% | 1.34% | <**0.001** | 0.95% | 1.10% | 0.14 | 0.99% | 1.51% | <**0.001** | 0.94% | 1.27% | <**0.001** |
| **Location of the Call** | | | | | | | | | | | | | | | |
| Urban (vs. rural areas)[3] | 79.74% | 80.88% | **0.002** | 80.07% | 78.29% | <**0.001** | 80.06% | 79.75% | 0.40 | 80.15% | 78.90% | <**0.001** | 80.01% | 79.50% | **0.007** |
| **Reason for Call** (as a proportion of all calls) | | | | | | | | | | | | | | | |
| Overdose/intoxication (total) | 4.13% | 4.51% | 0.04 | 5.03% | 6.83% | <**0.001** | 5.52% | 6.08% | **0.01** | 4.67% | 4.66% | 0.98 | 4.84% | 5.46% | <**0.001** |
| Overdose/intoxication (in rural areas) | 2.91% | 3.15% | 0.50 | 4.12% | 5.17% | **0.02** | 4.01% | 4.86% | 0.04 | 3.23% | 3.05% | 0.62 | 3.56% | 4.03% | 0.02 |
| Overdose/intoxication (in urban areas) | 4.44% | 4.83% | 0.07 | 5.27% | 7.27% | <**0.001** | 5.90% | 6.39% | 0.05 | 5.02% | 5.08% | 0.78 | 5.16% | 5.83% | <**0.001** |
| Behavioral (total) | 10.62% | 10.49% | 0.64 | 11.99% | 12.40% | 0.20 | 12.42% | 12.56% | 0.65 | 11.21% | 9.73% | <**0.001** | 11.56% | 11.24% | 0.03 |
| Behavioral (in rural areas) | 9.39% | 9.41% | 0.98 | 11.08% | 11.71% | 0.36 | 11.38% | 11.83% | 0.50 | 10.01% | 8.08% | **0.001** | 10.46% | 10.20% | 0.41 |
| Behavioral (in urban areas) | 10.94% | 10.73% | 0.52 | 12.21% | 12.57% | 0.33 | 12.68% | 12.75% | 0.85 | 11.51% | 10.21% | <**0.001** | 11.84% | 11.52% | 0.06 |
| Motor vehicle collision (total) | 3.32% | 2.18% | <**0.001** | 4.13% | 3.27% | <**0.001** | 4.29% | 3.66% | <**0.001** | 3.91% | 2.94% | <**0.001** | 3.91% | 3.00% | <**0.001** |
| Motor vehicle collision (in rural areas) | 4.05% | 2.62% | <**0.001** | 5.02% | 4.43% | 0.21 | 5.34% | 4.57% | 0.10 | 5.11% | 3.65% | **0.001** | 4.88% | 3.82% | <**0.001** |
| Motor vehicle collision (in urban areas) | 3.10% | 2.03% | <**0.001** | 3.68% | 2.93% | <**0.001** | 3.99% | 3.42% | **0.005** | 3.58% | 2.77% | <**0.001** | 3.63% | 2.78% | <**0.001** |

[1] Denominator is the total number of transports

[2] Denominator is total number of calls in the specified time frame.

[3] Rurality was missing in fewer than 0.2% calls

[4] P-values are as-reported, but are compared to a Bonferroni-corrected critical p-value for each category. P-values that achieve statistical significance after Bonferroni correction are indicated in **bold** font.

**A**

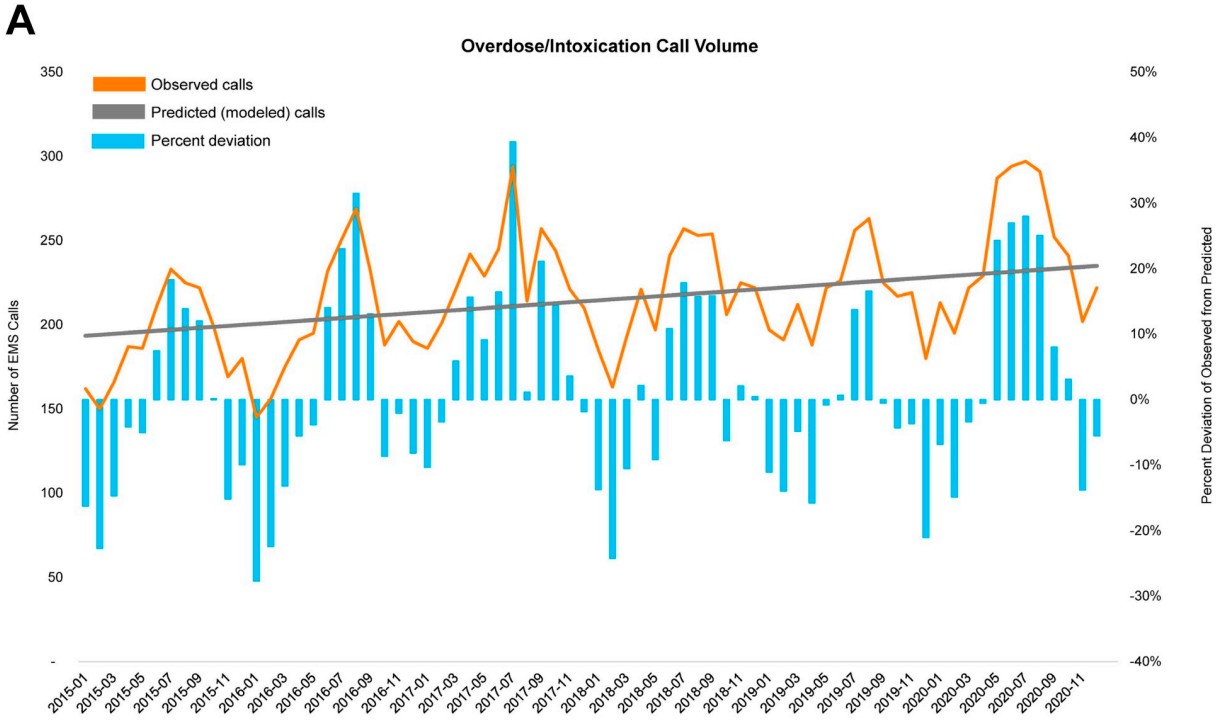

**B**

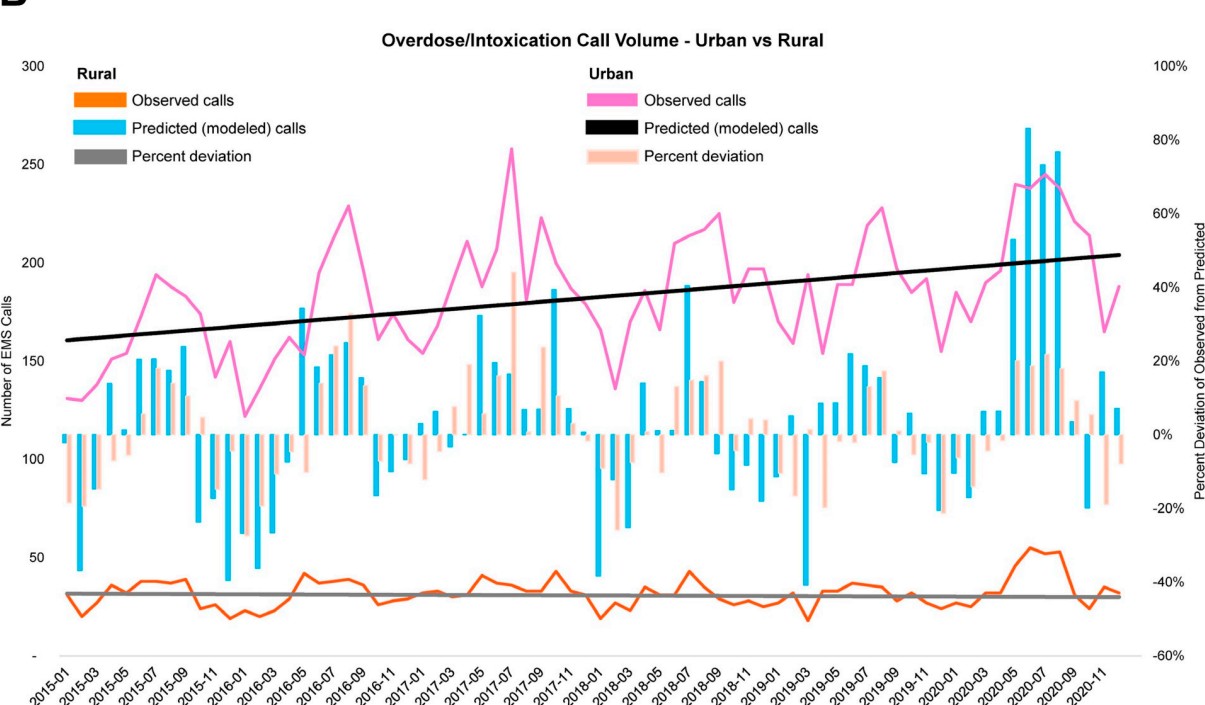

**Fig 3. Observed and predicted EMS calls for overdose/intoxication, 2015–2020.** Changes in call volumes per month for all regions included in the study, overall (upper graph) and subset by rural status of the call originating location (bottom graph). Observed call volumes were compared to those expected based on a linear model of event rates across the study period. Percent deviation is calculated as the observed volume minus the expected (i.e., modeled) volume, divided by the expected volume.

proportion of calls being for higher clinical severity, but a decreased number of calls overall and a higher transport rate, suggesting that patients in our study limited their EMS utilization more than those in Calgary and New Zealand, potentially because these studies focused on time periods with enforced lockdowns and higher COVID-19 case rates than our study. Conversely, lower rates of EMS utilization with the start of the COVID-19 pandemic were observed in other Canadian [32] and U.S. [33] studies, underscoring the different EMS utilization and health seeking behaviors of individuals both within and across countries.

Compared to urban areas, rural areas saw a smaller impact on EMS activation volume and severity. There are several potential explanations for this. It has been reported that in the beginning of the pandemic, COVID-19 impacted rural communities less than urban centers due to the virus arriving in cities first and the challenges to social distancing posed by high population density in urban areas [16, 34]. The difference in population-adjusted COVID-19 cases and mortality rates shifted as the pandemic progressed, with both fatality and incidence in rural areas surpassing those in urban areas by the end of 2020 [35–37]. Rural communities were found to be less likely to change their behavior, such as wearing masks, abiding by stay-at-home orders, or practicing social distancing, in response to the COVID-19 pandemic [37, 38]. Lack of nearby care and distance from definitive care necessitates the use of EMS for many patients in rural communities [26, 39]. Therefore, patients in rural areas may have been less likely to change their healthcare utilization patterns. It will be important to build upon our findings to examine differences in ED and hospital utilization in rural as compared to urban areas throughout the pandemic. EMS is also more utilized in urban areas. [40] with potential for a greater proportion of low acuity activations that could have been eliminated while rationing care in the setting of COVID-19. In contrast, rural areas rely more heavily on EMS for transportation to the ED and as a source of medical care than do urban populations, ensuring that EMS utilization does not decline despite potential external pressures [16, 17].

We also observed a decrease in EMS activations related to motor vehicle collisions. This observation is consistent with previous reports of decreased automobile traffic and accidents during the pandemic, as people were less likely to travel [21, 24, 41, 42]. On the other hand, we observed an overall increase in overdose or intoxication-related EMS activations during the COVID-19 pandemic, also consistent with emerging literature from other settings [23, 25]. Possible explanations include stress from lockdown restrictions, loss of employment, and uncertainties regarding the pandemic's course, all of which can lead to increased alcohol and substance use [22]. The increase in overdose or intoxication-related EMS activations was more pronounced in urban areas, reflecting the profound impact of the pandemic and associated economic and social changes on urban residents.

This study is strengthened by the availability of granular activation-level data from a multi-state ALS ambulance agency that serves both rural and urban areas. However, limitations do exist. Mayo Clinic Ambulance Service serves communities in the U.S. Upper Midwest, and our findings may not generalize to other areas of the U.S. While included service areas represent both rural and urban populations, the largest included cities have fewer than 120,000 residents. The impact of COVID-19 on EMS utilization in larger metropolitan areas may not be generalizable from these findings. However, data on the pandemic's impact on healthcare delivery in smaller, Midwest, and rural areas have been scarce, increasing the impact and significance of our findings.

The collected data are also limited to what is available in EMS patient care records and lacks patient-level clinical information and outcomes data, as would be available within a health system, because these data and services are outside the scope of EMS. Nevertheless, the use of activation-level EMS data of a sole ALS ambulance provider in the covered geographic areas ensures complete capture of EMS utilization by people living in this geography; such

capture may not be possible if relying on health system electronic health record data. This lack of comprehensive patient- and population-level data similarly limited the robustness of the model used to estimate the expected call volumes over the study period, which also did not consider seasonality, changing population demographics, and other temporal events. There is no gold standard for categorizing EMS activation severity, and we relied on the use of lights and sirens during transport to indicate EMS activations as higher acuity. However, the decision to use lights and sirens is influenced both by the objective severity of the patient's illness as well as by subjective factors such as the EMS providers' experience and comfort level managing the patient, resources available on scene, distance to the ED, and traffic conditions en route. Indeed, small differences in lights and sirens use were already apparent in Q1 of 2020, before the first cases of COVID-19 were detected in our region. Finally, EMS activation volumes presented here are not population-adjusted, but we accounted for this by considering temporal trends from the previous five-year period to compare EMS utilization trends from the same population.

## Conclusion

EMS activation volume and severity were affected by the COVID-19 pandemic. EMS activation volume decreased, particularly in urban areas, while severity of activations increased, suggesting deferral of lower acuity care and potentially delayed EMS activation. We observed a disproportionate increase in activations for overdose or intoxication, reflecting the increase in drug-related morbidity and mortality observed during the COVID-19 period [7]. We also observed a decrease in motor vehicle accidents, reflecting the decreased travel amongst stay-at-home orders during the COVID-19 period [37]. Further research is needed to fully understand the impact of deferred EMS care on patient outcomes.

## Supporting information

**S1 Checklist. STROBE statement—Checklist of items that should be included in reports of *cohort studies.***
(DOCX)

**S1 File.**
(XLSX)

## Author Contributions

**Conceptualization:** Lucas A. Myers, Rozalina G. McCoy.

**Data curation:** M. Carson Rogerson, IV.

**Formal analysis:** M. Carson Rogerson, IV.

**Funding acquisition:** Rozalina G. McCoy.

**Methodology:** Shuo J. Huang.

**Supervision:** Rozalina G. McCoy.

**Visualization:** Shuo J. Huang.

**Writing – original draft:** Moshe Shalom, Lucas A. Myers.

**Writing – review & editing:** Moshe Shalom, Brett Boggust, M. Carson Rogerson, IV, Lucas A. Myers, Shuo J. Huang, Rozalina G. McCoy.

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
