## [Decision Letter · Decision Letter 0]

2 Apr 2024

PONE-D-23-44051Impact of COVID-19 on Emergency Medical Services Utilization and Severity in the U.S. Upper MidwestPLOS ONE

Dear Dr. Boggust,

Thank you for submitting your manuscript to PLOS ONE. After careful consideration, we feel that it has merit but does not fully meet PLOS ONE’s publication criteria as it currently stands. Therefore, we invite you to submit a revised version of the manuscript that addresses the points raised during the review process.

We look forward to receiving your revised manuscript.

Kind regards,

Jobst Augustin, Associate Professor/Senior lecturer

Academic Editor

PLOS ONE

Journal Requirements:

"This effort was funded by the National Institute of Diabetes and Digestive and Kidney Diseases (NIDDK) of the National Institute of Health (NIH) National Institute of Diabetes and Digestive and Kidney Diseases (NIDDK) grant number R03DK127010."

**Additional Editor Comments:**

The manuscript deals with an important and current topic with interesting results. However, the manuscript needs to be substantially revised before it can be accepted. Please read the reviewers' comments carefully and implement them accordingly. Another point: the maps (Figure 2) show absolute values. Absolute values should only be used in choropleth maps in justified exceptional cases. Accordingly, relative values (e.g. per 1000 inhabitants) should normally be used. Please check this again.

Reviewers' comments:

Reviewer's Responses to Questions

**Comments to the Author**

1. Is the manuscript technically sound, and do the data support the conclusions?

Reviewer #1: Partly

Reviewer #2: Partly

2. Has the statistical analysis been performed appropriately and rigorously? 

Reviewer #1: I Don't Know

Reviewer #2: Yes

3. Have the authors made all data underlying the findings in their manuscript fully available?

Reviewer #1: No

Reviewer #2: Yes

4. Is the manuscript presented in an intelligible fashion and written in standard English?

Reviewer #1: Yes

Reviewer #2: Yes

5. Review Comments to the Author

**Reviewer #1: **The authors provide valuable insights into the first phase of the COVID-19 pandemic by analyzing 911 calls made to Mayo Clinic ambulance services. They use appropriate descriptive methods to describe utilization in 2020 and to compare it to previous years and the modeled number of expected calls in 2020. They find that utilization declines, especially during the first months when the pandemic hits the US (March-June). The sub-analysis of calls due to motor vehicle collisions follows this underlying trend, whereas calls because of overdose/intoxication increase, and results for the category “behavioral” are mixed. Declines were more pronounced in urban areas. There is also an increased use of lights and sirens, as well as an increase in calls with transport and with death prior to arrival, however, these differences are rather small.

The manuscript is well-written and it is easy to follow the structure. The authors followed the STROBE reporting guidelines. However, there are a few concerns that I believe should be addressed. This applies in particular to the analysis and presentation in the tables as well as the graphical representation and the definition of the "pandemic period.” My detailed comments are as follows:

Major Issues

Line 113: Please clarify what “14 locations” refers to. Are these distinct geographic areas or communities? Or hospitals as the main destination of transport? Can you indicate how many other operators cover the same area as Mayo Clinic Ambulance? This would make it easier to set this into context for readers that are not familiar with the US ambulance system.

I am no statistical expert, so perhaps I am not correctly assessing the authors' approach to the analysis. However, I'm confused about why, when comparing distributions, there are multiple Chi-Square test p-values reported for variables that have more than two categories and add up to 100%. This is seen in Table 1 for “patient age” and Table 2 for “call outcome.” This is not the case for categories “Gender” and “Rurality” in Table 1, where the authors only report one p-value.

I know that by convention, a p-value higher than 0.05 is often used to determine “significance” (i.e., statistical significance). However, classifying results as “significant” or not based on a p-value cutoff is arbitrary, and results might still be inconclusive (for further explanation see e.g.: Cochrane Handbook for Systematic Reviews of Interventions. Chapter 15: Interpreting results and drawing conclusions. 15.3.2 P values and statistical significance). Many of the observed “statistically significant” differences are also quite small (e.g., in Table 1: 93.99% vs 94.96% (p>0.001) transport vs no transport Q1). Therefore, I recommend not using this wording and removing the sentence.

Can the authors also explain why Bonferroni correction is necessary? I am also not sure whether Bonferroni correction was actually applied, as Footnote 4 of Table 2 says that “p-values are as reported”, and that the results were only compared to Bonferroni corrected values. Which p-value is reported, and what were the results of the comparison?

In line 156, the authors describe that missing values are presented as a separate category (unknown). Does that mean that, in Table 2, there was only one variable (Location of call) with missing values (0.17% pre-COVID, 0.29% post-COVID), and all other variables did not? Footnote 3 of Table 2: Does the sentence “was not taken into consideration” imply complete case analysis? Could the authors please specify?

Why did the authors choose to display calls from 2019-2020 only? Data are available from 2015. Displaying these data would be helpful to further separate random variations in the number of calls from variation caused by COVID. If this is because the presentation of all years would take up too much space, I suggest basing the “comparison” not only on 2019 but on the average of previous years, as was done in the tables (pre-COVID).

In line 189, the authors describe percentage changes (6.8% / 0.05%) and refer to Figure 1. However, I am unable to derive percentage changes from the figure. I recommend reporting the absolute number that can be derived from the graph and adding the percentage change in brackets.

I think it is useful to compare the actual number of calls to the projected number of calls. However, the model used to project call volume is very simple and likely does not represent reality (e.g., does not take into consideration seasonality, demographics, events, etc.). I suggest mentioning in the limitations that the model is also only a guess of what the actual number of calls would have been without COVID.

In Figure 1, labels for the left and right axes are missing. The legend is difficult to understand. Please specify the axes and the variable names. The result for the variable “yearmon” seems to be a straight line on the x-axis. This is probably a mistake, which also shifts the label of axis ticks on the x-axis and makes it hard to understand which data point belongs to which month. In general, it is hard to evaluate the graphs as long as important information about interpretation (axes labeling) is missing and axis ticks of the left and right axes are not aligned. Labels of the left and right axes also need to be added to Figure 3, as well as the legend of the first part of Figure 3.

I agree with the conclusion that the severity of calls, based on the use of lights and sirens and calls with fatal outcomes, increased. However, the difference is quite small and this is already observed in Q1 (see Table 2), where the upcoming pandemic likely did not impact people's decisions to call an ambulance. This should be mentioned early in the discussion. In general, defining the year 2020 as the “COVID-19 Period” might be misleading. Even though the pandemic began in late 2019, it can be assumed that it still had little impact on people in the USA in the first quarter of 2020. To avoid any misunderstanding, I suggest changing the column header of Table 1 to “2015-2019” or “pre-2020” and “2020”, as is the case in Table 2. I would also recommend rephrasing this wording in the abstract and the rest of the manuscript, wherever the authors refer to results that are derived from data of the whole year of 2020. Alternatively, the analyses could be adjusted so that the pandemic period only covers Q2-Q4.

I would appreciate it if the authors added a short paragraph where they describe the differences between the different pandemic stages in 2020 in a bit more detail. There are notable differences between different quarters and months of the year.

Line 270: How do the authors explain that others found lower rates of transport (e.g., Lane et al (DOI: https://doi.org/10.9778/cmajo.20200313, Dicker et al: https://doi.org/10.1136/bmjopen-2020-044726, Hegenberg et al doi: 10.1186/s13049-019-0607-5) whereas the amount of transported patients was increasing in their study? Is this explained by different EMS systems?

Minor issues:

• Abstract: please specify which region the figure (1 million lives) refers to.

• Line 85: this sentence might be there due to a longer submission process, yet I would say that, gladly, the COVID-19 pandemic is over. It is still important to learn from these events, please adapt the sentence.

• Line 130: There is a word missing (of) after “combination.”

• Table 2: When reporting “Need for CPR”: for better readability, I would suggest formatting this variable so that it can be quickly recognized that rural vs urban is a stratification. The same applies to the variable “reason for call.”

• Line 324: the historic comparison is not exactly the same as adjustment. If there are changes in population numbers, the population is not the same. I assume that this is not the case to a large extent, however, this should be added to the limitations.

• The data availability statement says that all data are available without restriction. I am not sure whether the respective method of retrieving these data (repository etc.) needs to be included.

• The resolution of the map could be better; it is a bit difficult to read. This will probably be changed during the publication process.

**Reviewer #2: **This manuscript compares EMS call volumes in the five years prior to the COVID-19 pandemic (2015-1019) with the year of the pandemic (2020). In addition to the number of calls, the demographics, type and severity of the emergency call are compared and an urban-rural comparison is made. Overall, a decrease in the number of calls was observed, although the severity increased. During the pandemic, older people and people in rural areas were more likely to call 911.

The manuscript is well written overall, but there are still a few details to get a better understanding of the methodology and interpretation.

It is not clear to me where the expected values for 2020 have been used. Do they only refer to the figures?

I also wondered, if the data is available on a monthly basis, why 2020 is summarized as the "pandemic year" and not the actual start of the pandemic.

For the comparisons, 95% confidence intervals would be useful instead of the many p-values or additionally. The p-value depends on the number of cases, which is very large here. Confidence intervals therefore make sense in order to better capture the differences.

The figures as a whole should be revised again.

I have the following questions and comments:

- Fig. 1: A more detailed description of the figures would help to better understand the graph. Is it possible that the x-axis has slipped? A better labeling would also be useful here, it seems as if the variable names have not been adjusted (e.g.: yearmon)

- Fig. 2: The legend describes that the 2018 population is mapped. However, the number of calls can be seen here. What is meant by "Population data are mapped by ZIP code"? How were the 5 years of the Prior-2020 period summarized?

- Fig. 3: Labeling of the upper graph is missing

- l. 174: The goodness of fit parameters belong in the results section rather than the methods section

- l. 186-188: The monthly average number over the entire period is considered and a rate of change is determined from this, but here the average monthly rate before the pandemic should be compared with the year during the pandemic.

- l. 163 How were the 5 years combined?

- I. 165 "number of calls" - this cannot be the absolute number of the 5 years, is it an averaged value?

- l. 178: Were the expected values used from here on or the observed values?

- l. 194 and Table 1: Why was age used here as a continuous variable and then age groups were compared again in Table 1? Have the p-values been adjusted for age? Overall, too much was tested here. Especially for Table 1, 95% confidence intervals would be useful.

- Table 2: Are these observed or expected values? Strictly speaking, Q1 2020 also belongs to the "pre-2020" period. Significant differences can also be observed here. Can this therefore indicate a difference at all that relates to the pandemic? Here, too, 95% confidence intervals would be helpful.

6. PLOS authors have the option to publish the peer review history of their article (what does this mean?). If published, this will include your full peer review and any attached files.

Reviewer #1: No

Reviewer #2: No

---

## [Author Response · Author response to Decision Letter 0]

3 Jul 2024

RESPONSE TO THE EDITOR---------------------------------------

The maps (Figure 2) show absolute values. Absolute values should only be used in choropleth maps in justified exceptional cases. Accordingly, relative values (e.g. per 1000 inhabitants) should normally be used. Please check this again.

RESPONSE: We are grateful for the Editor’s and Reviewers’ recommendations on improving this figure. We have revised Figure 2 to reflect the 2020 change, as compared to the annualized average from 2015-2019, in EMS call volumes per 10,000 of each county’s population in 2020.

RESPONSE TO REVIEWER 1-------------------------------------

1. Line 113: Please clarify what “14 locations” refers to. Are these distinct geographic areas or communities? Or hospitals as the main destination of transport? Can you indicate how many other operators cover the same area as Mayo Clinic Ambulance? This would make it easier to set this into context for readers that are not familiar with the US ambulance system.

RESPONSE: We appreciate the opportunity to clarify this information, which can help contextualize study findings. Mayo Clinic Ambulance is a large multi-state ambulance service that serves communities across Minnesota and Wisconsin; the “14 sites” referred to primary service hubs, but for greater clarity we rephrased this section to list the specific areas served by Mayo Clinic Ambulance in each state. We further clarify that only one ambulance service can have its primary service area in a given location, such that there are no other operators that cover the same geographic area.

[Page 7] Mayo Clinic Ambulance is an ALS provider and the primary response, treatment, and ground and air medical transport service in communities across southern, central, and western Minnesota (hubs in Albert Lea, Austin, Duluth, Fairmont, Litchfield, Little Falls, Mankato, Owatonna, Plainview, Rochester, and St. Cloud) as well as western Wisconsin (hubs in Barron, Eau Claire, Osseo, and Superior), covering 6,894 square miles of urban, suburban, and rural areas. Mayo Clinic Ambulance is staffed by paramedics and emergency medical technicians, and responds to approximately 100,000 requests for service annually, including 75,000 emergent 911 calls, excluding calls for inter-facility transport. In Minnesota and Wisconsin, a single ambulance service is licensed to provide emergency care to an assigned primary service area with defined geographic boundaries. 

2. I am no statistical expert, so perhaps I am not correctly assessing the authors' approach to the analysis. However, I'm confused about why, when comparing distributions, there are multiple Chi-Square test p-values reported for variables that have more than two categories and add up to 100%. This is seen in Table 1 for “patient age” and Table 2 for “call outcome.” This is not the case for categories “Gender” and “Rurality” in Table 1, where the authors only report one p-value.

RESPONSE: Thank you for calling our attention to this lack of clarity. For variables that have more than two categories (e.g., age in Table 1, call characteristics in Table 2), we compared across each category to better understand the differences between the 2015-2019 and 2020 time periods. For variables that have only two categories (e.g., rurality, gender), we compared across the entire variable as no additional information would be provided by comparing across each category. 

3. I know that by convention, a p-value higher than 0.05 is often used to determine “significance” (i.e., statistical significance). However, classifying results as “significant” or not based on a p-value cutoff is arbitrary, and results might still be inconclusive (for further explanation see e.g.: Cochrane Handbook for Systematic Reviews of Interventions. Chapter 15: Interpreting results and drawing conclusions. 15.3.2 P values and statistical significance). Many of the observed “statistically significant” differences are also quite small (e.g., in Table 1: 93.99% vs 94.96% (p>0.001) transport vs no transport Q1). Therefore, I recommend not using this wording and removing the sentence.

RESPONSE: We agree with the importance of differentiating between clinical and statistical significance. Per Reviewer’s recommendation, we revised the text throughout the manuscript to reflect this.

4. Can the authors also explain why Bonferroni correction is necessary? I am also not sure whether Bonferroni correction was actually applied, as Footnote 4 of Table 2 says that “p-values are as reported”, and that the results were only compared to Bonferroni corrected values. Which p-value is reported, and what were the results of the comparison?

RESPONSE: Bonferroni correction was necessary when assessing outcomes because our analysis included multiple comparisons; the goal of Bonferroni correction is to reduce risk of Type 1 error. We clarify this in the Methods. In Table 2, we adjusted the p-value level that would signify statistical significance after Bonferroni correction. We apologize that the bold font we had intended to use in Table 2 to signify statistical significance did not come across in the original submission; it is included now in the revision.

5. In line 156, the authors describe that missing values are presented as a separate category (unknown). Does that mean that, in Table 2, there was only one variable (Location of call) with missing values (0.17% pre-COVID, 0.29% post-COVID), and all other variables did not? Footnote 3 of Table 2: Does the sentence “was not taken into consideration” imply complete case analysis? Could the authors please specify?

RESPONSE: The Reviewer is correct, that only the location of the call included missing data, while all other outcome variables were complete without missingness. When comparing across categories using the chi-square test, any missing values are ignored by SAS and only the distribution of non-missing categories are considered. Because this is the standard mode of analysis that is inferred from the specified methods used, we deleted this phrase from the table legend to avoid confusion.

6. Why did the authors choose to display calls from 2019-2020 only? Data are available from 2015. Displaying these data would be helpful to further separate random variations in the number of calls from variation caused by COVID. If this is because the presentation of all years would take up too much space, I suggest basing the “comparison” not only on 2019 but on the average of previous years, as was done in the tables (pre-COVID).

RESPONSE: We apologize for this oversight and corrected Figures 1 and 3 to include the full 2015-2020 study period and clearly label the axes and legends.

7. In line 189, the authors describe percentage changes (6.8% / 0.05%) and refer to Figure 1. However, I am unable to derive percentage changes from the figure. I recommend reporting the absolute number that can be derived from the graph and adding the percentage change in brackets.

RESPONSE: We have included the absolute numbers of expected and observed call volumes to the results section. 

8. I think it is useful to compare the actual number of calls to the projected number of calls. However, the model used to project call volume is very simple and likely does not represent reality (e.g., does not take into consideration seasonality, demographics, events, etc.). I suggest mentioning in the limitations that the model is also only a guess of what the actual number of calls would have been without COVID.

RESPONSE: We appreciate this feedback and added the following limitation to the Discussion:

[Page 20] This lack of comprehensive patient- and population-level data similarly limited the robustness of the model used to estimate the expected call volumes over the study period, which also did not consider seasonality, changing population demographics, and other temporal events.

9. In Figure 1, labels for the left and right axes are missing. The legend is difficult to understand. Please specify the axes and the variable names. The result for the variable “yearmon” seems to be a straight line on the x-axis. This is probably a mistake, which also shifts the label of axis ticks on the x-axis and makes it hard to understand which data point belongs to which month. In general, it is hard to evaluate the graphs as long as important information about interpretation (axes labeling) is missing and axis ticks of the left and right axes are not aligned. Labels of the left and right axes also need to be added to Figure 3, as well as the legend of the first part of Figure 3.

RESPONSE: We apologize for this oversight in our submission and corrected Figures 1 and 3 to clearly label the axes and legends.

10. I agree with the conclusion that the severity of calls, based on the use of lights and sirens and calls with fatal outcomes, increased. However, the difference is quite small and this is already observed in Q1 (see Table 2), where the upcoming pandemic likely did not impact people's decisions to call an ambulance. This should be mentioned early in the discussion. In general, defining the year 2020 as the “COVID-19 Period” might be misleading. Even though the pandemic began in late 2019, it can be assumed that it still had little impact on people in the USA in the first quarter of 2020. To avoid any misunderstanding, I suggest changing the column header of Table 1 to “2015-2019” or “pre-2020” and “2020”, as is the case in Table 2. I would also recommend rephrasing this wording in the abstract and the rest of the manuscript, wherever the authors refer to results that are derived from data of the whole year of 2020. Alternatively, the analyses could be adjusted so that the pandemic period only covers Q2-Q4.

RESPONSE: We appreciate the Reviewer’s feedback and recommendations. Accordingly, we revised our language to clearly delineate statistical vs. clinical significance and to refer to the two examined time periods as 2015-2019 and 2020 throughout the text and in the tables.

[Pages 20-21] Indeed, small differences in lights and sirens use were already apparent in Q1 of 2020, before the first cases of COVID-19 were detected in our region.

11. I would appreciate it if the authors added a short paragraph where they describe the differences between the different pandemic stages in 2020 in a bit more detail. There are notable differences between different quarters and months of the year.

RESPONSE: We are happy to provide additional context and detail about how the COVID-19 pandemic began and proceeded throughout 2020 across Minnesota and Wisconsin. We specifically added the following text in different areas of the Methods section:

[Page 6] We used Mayo Clinic Ambulance PCR to identify all EMS encounters between January 1, 2015 and December 31, 2020, subsetting this period to 2015-2019 (baseline, pre-pandemic period) and 2020. First cases of COVID-19 were detected in Wisconsin on February 5, 2020 and in Minnesota on March 6, 2020.[19] Stay-at-home orders started on March 25 and March 28, respectively, while schools closed on March 18 in both states.[19] COVID-19 deaths first began to peak in early October 2020 in Wisconsin, while in Minnesota there was a smaller mortality spike in May 2020 followed by a larger peak in November/December 2020.[20]

[Page 9] Calendar quarters were defined as three-month periods beginning January 1 of a given calendar year. This approach was chosen to standardize time periods of outcome ascertainment across the two states included in our analysis, which had different COVID-19 case and fatality trajectories and policy approaches to risk mitigation.

12. Line 270: How do the authors explain that others found lower rates of transport (e.g., Lane et al (DOI: https://doi.org/10.9778/cmajo.20200313, Dicker et al: https://doi.org/10.1136/bmjopen-2020-044726, Hegenberg et al doi: 10.1186/s13049-019-0607-5) whereas the amount of transported patients was increasing in their study? Is this explained by different EMS systems?

RESPONSE: We appreciate the Reviewer calling our attention to these important studies. We added a paragraph to the discussion addressing these and several other pertinent studies we have identified in the process of an updated literature review.

[Page 18] Our findings contrast with other studies that examined EMS utilization in the context of the COVID-19 pandemic in other settings and populations. Lane et al examined EMS utilization in Calgary, Canada between December 2019 and June 2020 (corresponding roughly to Q1-Q2 2020 in our study), with the first case of COVID-19 reported midway through that time period.[25] They found that in 2020, as compared to 2017-2019, overall EMS call volumes increased, as did the proportion of calls that were of higher clinical severity, while overall transport rates decreased (observed mainly in the lower severity calls).[25] Higher EMS volumes and greater proportion of calls for low-acuity needs were also observed in a New Zealand study by Dicker et al, which examined EMS utilization in March and April 2020 during a peak of infection rates and a general country lockdown period.[26] We, too, found a higher proportion of calls being for higher clinical severity, but a decreased number of calls overall and a higher transport rate, suggesting that patients in our study limited their EMS utilization more than those in Calgary and New Zealand, potentially because these studies focused on time periods with enforced lockdowns and higher COVID-19 case rates than our study. Conversely, lower rates of EMS utilization with the start of the COVID-19 pandemic were observed in other Canadian[27] and U.S.[28] studies, underscoring the different EMS utilization and health seeking behaviors of individuals both within and across countries. 

13. Abstract: please specify which region the figure (1 million lives) refers to.

RESPONSE: The abstract and introduction have been modified to specify that the over one million lives lost to the COVID-19 pandemic was referring to the United States.

14. Line 85: this sentence might be there due to a longer submission process, yet I would say that, gladly, the COVID-19 pandemic is over. It is still important to learn from these events, please adapt the sentence.

RESPONSE: We share the Reviewer’s sentiment in this regard and updated this sentence to read as follows:

[Page 5] With the COVID-19 pandemic now mostly behind us, it is important to understand the impact of the pandemic – in its different phases – on EMS utilization.

15. Line 130: There is a word missing (of) after “combination.”

RESPONSE: This was corrected.

16. Table 2: When reporting “Need for CPR”: for better readability, I would suggest formatting this variable so that it can be quickly recognized that rural vs urban is a stratification. The same applies to the variable “reason for call.”

RESPONSE: We have reformatted Table 2 for ease of readability. We anticipate that further visual and editorial changes will be made during the copy-editing process to adhere to journal guidelines.

17. Line 324: the historic comparison is not exactly the same as adjustment. If there are changes in population numbers, the population is not the same. I assume that this is not the case to a large extent, however, this should be added to the limitations.

RESPONSE: We appreciate the Reviewer’s comment and confirmed that we do not suggest that our analyses were adjusted in the text or any of the table or figure legends. We also added the following text to the Discussion:

[Page 20] This lack of comprehensive patient- and population-level data similarly limited the robustness of the model used to estimate the expected call volumes over the study period, which also did not consider seasonality, changing population demographics, and other temporal events.

18. The data availability statement says that all data are available without restriction. I am not sure whether the respective method of retrieving these data (repository etc.) needs to be included.

RESPONSE: We apologize for the error made during the manuscript submission process. Data are not available due to restrictions on data s

---

## [Editor Report · Decision Letter 1]

13 Aug 2024

PONE-D-23-44051R1Impact of COVID-19 on Emergency Medical Services Utilization and Severity in the U.S. Upper MidwestPLOS ONE

Dear Dr. McCoy,

Thank you for submitting your manuscript to PLOS ONE. After careful consideration, we feel that it has merit but does not fully meet PLOS ONE’s publication criteria as it currently stands. Therefore, we invite you to submit a revised version of the manuscript that addresses the points raised during the review process.

**
*Please revise the figures as they are at least partially confusing. This applies in particular to figures 2 and 5. Solid lines can also be shown in combination with dashed lines. Also make sure that the colours are clearly distinguishable in the printed format and that readers with colour vision problems (e.g. red/green) can read the figure. *
**

We look forward to receiving your revised manuscript.

Kind regards,

Jobst Augustin, Associate Professor/Senior lecturer

Academic Editor

PLOS ONE
---

## [Author Response · Author response to Decision Letter 1]

19 Aug 2024

I revised figures 1 and 3 (both panels of each) to avoid use of red/green colors, and to hopefully make all the colors easier to see. I am happy to also share a PowerPoint version of all the figures for the journal editorial team to manipular the colors per their expertise, if that would be helpful. I also uploaded a final clean version of the manuscript as no changes to the text were required.

---

## [Editor Report · Decision Letter 2]

23 Aug 2024

Impact of COVID-19 on Emergency Medical Services Utilization and Severity in the U.S. Upper Midwest

PONE-D-23-44051R2

Dear Dr. McCoy,

We’re pleased to inform you that your manuscript has been judged scientifically suitable for publication and will be formally accepted for publication once it meets all outstanding technical requirements.

Kind regards,

Jobst Augustin, Associate Professor/Senior lecturer

Academic Editor

PLOS ONE

---

## [Editor Report · Acceptance letter]

20 Sep 2024

PONE-D-23-44051R2 

PLOS ONE

Dear Dr. McCoy, 

I'm pleased to inform you that your manuscript has been deemed suitable for publication in PLOS ONE. Congratulations! Your manuscript is now being handed over to our production team.

Kind regards, 

on behalf of

Dr. Jobst Augustin 

Academic Editor

PLOS ONE